# STOCHASTIC ADVERSARIAL VIDEO PREDICTION

## ABSTRACT

Being able to predict what may happen in the future requires an in-depth understanding of the physical and causal rules that govern the world. A model that is able to do so has a number of appealing applications, from robotic planning to representation learning. However, learning to predict raw future observations, such as frames in a video, is exceedingly challenging—the ambiguous nature of the problem can cause a naively designed model to average together possible futures into a single, blurry prediction. Recently, this has been addressed by two distinct approaches: (a) latent variational variable models that explicitly model underlying stochasticity and (b) adversarially-trained models that aim to produce naturalistic images. However, a standard latent variable model can struggle to produce realistic results, and a standard adversarially-trained model underutilizes latent variables and fails to produce diverse predictions. We show that these distinct methods are in fact complementary. Combining the two produces predictions that look more realistic to human raters and better cover the range of possible futures. Our method outperforms prior works in these aspects.

## 1 INTRODUCTION

When we interact with objects in our environment, we can easily imagine the consequences of our actions: push a ball and it will roll; drop a vase and it will break. The ability to imagine future outcomes provides an appealing avenue for learning about the world. Unlabeled video sequences can be gathered autonomously with minimal human intervention, and a machine that learns to predict future events will gain an in-depth and functional understanding of its environment. This leads naturally to the problem of video prediction—given a sequence of context frames, and optionally a proposed action sequence, generate the pixels of the future frames. Once trained, such a model could be used to determine which actions can bring about desired outcomes (Finn et al., 2016; Ebert et al., 2017). Unfortunately, accurate and naturalistic video prediction remains an open problem.

One major challenge in video prediction is the ambiguous nature of the problem. While frames in the immediate future can be extrapolated with high precision, the space of possibilities diverges beyond a few frames, and the problem becomes multimodal by nature. Methods that use deterministic models and loss functions unequipped to handle this inherent uncertainty, such as mean-squared error (MSE), will average together possible futures, producing blurry predictions. Prior works have explored stochastic models for video prediction (Babaeizadeh et al., 2018; Denton & Fergus, 2018), using the framework of variational autoencoders (VAEs) (Kingma & Welling, 2014). These models predict possible futures by sampling latent variables. During training, they optimize a variational lower bound on the likelihood of the data in a latent variable model. However, the posterior is still a pixel-wise MSE loss, corresponding to the log-likelihood under a fully factorized Gaussian distribution. This makes training tractable, but causes them to still make blurry and unrealistic predictions when the latent variables alone do not adequately capture the uncertainty.

Another relevant branch of recent work has been generative adversarial networks (GANs) (Goodfellow et al., 2014) for image generation. Here, a generator network is trained to produce images that are indistinguishable from real images, under the guidance of a learned discriminator network trained to classify images as real or generated. The discriminator operates on patches or entire images, and is thus capable of modeling the joint distribution of pixels. Although this overcomes the limitations of pixel-wise losses, GANs are notoriously susceptible to mode collapse, where latent random variables are often ignored by the model, especially in the conditional setting. This makes them difficult to apply to generation of diverse and plausible futures, conditioned on context frames.

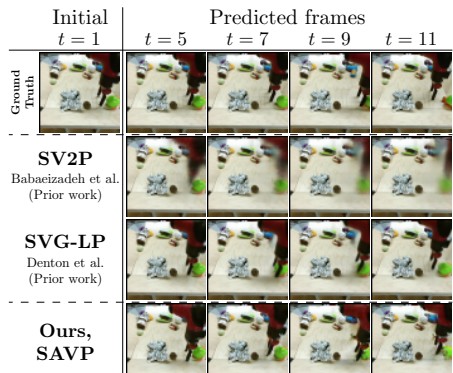 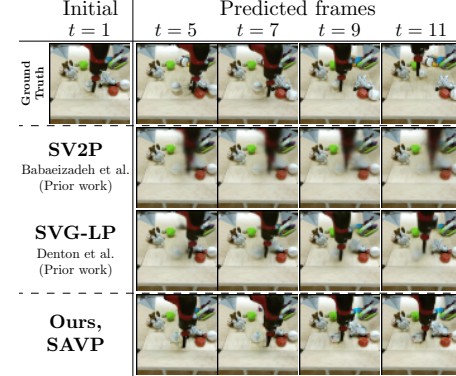

Figure 1: **Example results.** While the SV2P method (Babaeizadeh et al., 2018) produces blurry images, our method maintains sharpness and realism through time. The prior SVG-LP method (Denton & Fergus, 2018) produces sharper predictions, but still blurs out objects in the background (left) or objects that interact with the robot such as the baseball (right).

To address these challenges, we propose a model that combines both adversarial losses and latent variables to enable realistic stochastic video prediction. Our model consists of a video prediction network that can sample multiple plausible futures by sampling time-varying stochastic latent variables and decoding them into multiple frames. At training time, an inference network estimates the distribution of these latent variables, and video discriminator networks classify generated videos from real. The full training objective is the variational lower bound used in VAEs combined with the adversarial loss used in GANs. This enables us to capture stochastic posterior distributions of videos while also modeling the spatiotemporal joint distribution of pixels. VAEs with no adversarial losses are also capable of modeling joint distributions, provided that the generator model doesn't assume any factorization of the pixels. This is the case of pixel-autoregressive models, such as Pixel Video Networks (Kalchbrenner et al., 2017), though training and inference with these models are impractically slow. In this work, we take a different approach and we instead focus on the losses.

The primary contribution of our work is an stochastic video prediction model based on VAE-GANs. To our knowledge, this is the first stochastic video prediction model that combines an adversarial loss with a latent variable model trained via the variational lower bound. Our experiments show that the VAE component greatly improves the diversity of the generated images, while the adversarial loss attains prediction results that are substantially more realistic than state-of-the-art methods, as shown in Fig. 1. We further present a comparison of various types of prediction models and losses, including VAE, GAN, and VAE-GAN models, and analyze the impact of these choices on prediction realism, diversity, and accuracy.

## 2 RELATED WORK

Recent developments in expressive generative models based on deep networks has led to impressive developments in video generation and prediction. Earlier approaches to prediction focused on models that generate pixels directly from the latent state of the model using both feed-forward (Ranzato et al., 2014; Mathieu et al., 2016) and recurrent (Oh et al., 2015; Xingjian et al., 2015) architectures. An alternative to generating pixels is transforming them by applying a constrained geometric distortion to a previous frame (Finn et al., 2016; De Brabandere et al., 2016; Xue et al., 2016; Byravan & Fox, 2016; Vondrick & Torralba, 2017; van Amersfoort et al., 2017; Liu et al., 2017; Chen et al., 2017; Lu et al., 2017; Walker et al., 2015; 2016; Liang et al., 2017). Aside from the design of the generator architecture, performance is strongly affected by the training objective. Simply minimizing MSE loss can lead to strong results for deterministic synthetic videos (Oh et al., 2015; Chiappa et al., 2017). However, on real-world videos that contain uncertainty, this loss can result in blurry predictions, as the model averages futures to avoid incurring a large MSE loss (Mathieu et al., 2016).

Incorporating uncertainty is critical for addressing this issue. One approach is to model the full joint distribution using pixel-autoregressive models (van den Oord et al., 2016; Kalchbrenner et al., 2017; Reed et al., 2017), though training and inference are impractically slow. Another approach is to train a latent variable model, such as in variational autoencoders (VAEs) (Kingma & Welling, 2014). Conditional VAEs have been used for prediction of optical flow trajectories (Walker et al.,

2016), single-frame prediction (Xue et al., 2016), and recently for stochastic multi-frame video prediction (Babaeizadeh et al., 2018; Denton & Fergus, 2018). While these models can model distributions over possible futures, the prediction distribution is still fully factorized over pixels, which still tends to produce blurry predictions.

Adversarial losses (Goodfellow et al., 2014) for image generation can produce substantially improved realism. However, these networks tend to be difficult to train and are susceptible to mode collapse. A number of prior works have used adversarial losses for deterministic video prediction (Mathieu et al., 2016; Vondrick & Torralba, 2017; Villegas et al., 2017; Lu et al., 2017; Zhou & Berg, 2016; Bhattacharjee & Das, 2017). Several prior works have also sought to produce unconditioned video generations (Vondrick et al., 2016; Saito et al., 2017; Tulyakov et al., 2018) and conditional generation with input noise (Chen et al., 2017; Tulyakov et al., 2018; Wang et al., 2018). We show that a GAN model with input noise can indeed generate realistic videos, but fails to adequately cover the space of possible futures. In contrast, our method, which combines latent variable models with an adversarial loss, produces videos that are both visually plausible and diverse.

Prior works have combined VAEs and GANs to produce stochastic and realistic predictions. Walker et al. (2017) predicts videos of humans by decomposing the problem into a VAE that predicts stochastic future poses and a GAN that generates videos conditioned on those poses and an image. VAE-GANs, which jointly optimize the VAE and GAN losses, have shown promising results for unconditional and conditional image generation (Larsen et al., 2016; Bao et al., 2017; Zhu et al., 2017), but have not been applied to video prediction. The video prediction setting presents two important challenges. First, conditional image generation can handle large appearance changes between the input and output, but suffer when attempting to produce large spatial changes. The video prediction setting is precisely the opposite—the appearance remains largely the same from frame to frame, but the most important changes are spatial. Secondly, video prediction involves sequential prediction, where it's increasingly difficult to predict farther into the future. Our approach is the first to use VAE-GANs in a recurrent setting for stochastic video prediction.

## 3 VIDEO PREDICTION WITH STOCHASTIC ADVERSARIAL MODELS

Our goal is to learn a stochastic video prediction model that can predict videos that are diverse and perceptually realistic, and where all predictions are plausible futures for the given initial image. In practice, we use a short initial sequence of images (typically two frames), though we will omit this in our derivation for ease of notation. Our model consists of a recurrent generator network $G$, which is a deterministic video prediction model that maps an initial image $\mathbf{x}_0$ and a sequence of latent random codes $\mathbf{z}_{0:T-1}$, to the predicted sequence of future images $\hat{\mathbf{x}}_{1:T}$. Intuitively, the latent codes encapsulate any ambiguous or stochastic events that might affect the future. At test time, we sample videos by first sampling the latent codes from a prior distribution $p(\mathbf{z}_t)$, and then passing them to the generator. We use a fixed unit Gaussian prior, $\mathcal{N}(0, 1)$. The training procedure for this includes elements of variational inference and generative adversarial networks. Before describing the training procedure, we formulate the problem in the context of VAEs and GANs.

### 3.1 VARIATIONAL AUTOENCODERS

Our recurrent generator predicts each frame given the previous frame and a random latent code. The previous frame passed to the generator is denoted as $\tilde{\mathbf{x}}_{t-1}$ to indicate that it could be a ground truth frame $\mathbf{x}_{t-1}$ (for the initial frames) or the last prediction $\hat{\mathbf{x}}_{t-1}$. The generator specifies a distribution $p(\mathbf{x}_t|\mathbf{x}_{t-1}, \mathbf{z}_{t-1})$, parametrized as a fixed-variance Laplacian distribution with mean $\hat{\mathbf{x}}_t = G(\mathbf{x}_{t-1}, \mathbf{z}_{t-1})$. The likelihood of the data $p(\mathbf{x}_{1:T}|\mathbf{x}_0)$ cannot be directly maximized, since it involves marginalizing over the latent variables, which is intractable in general. Thus, we instead maximize the variational lower bound of the log-likelihood. We approximate the posterior with a recognition model $q(\mathbf{z}_t|\mathbf{x}_{t:t+1})$, which is parametrized as a conditionally Gaussian distribution $\mathcal{N}(\mu_{\mathbf{z}_t}, \sigma^2_{\mathbf{z}_t})$, represented by a network $E(\mathbf{x}_{t:t+1})$. The encoder $E$ is conditioned on adjacent frames $\mathbf{x}_t$ and $\mathbf{x}_{t+1}$ in order to have temporally local latent variables $\mathbf{z}_t$ that capture the ambiguity for only that transition, a sensible choice when using independent and identically distributed Gaussian priors. Another choice is to use temporally correlated latent variables, which would require a stronger prior (e.g. as in Denton & Fergus (2018)). For simplicity, we opted for the former.

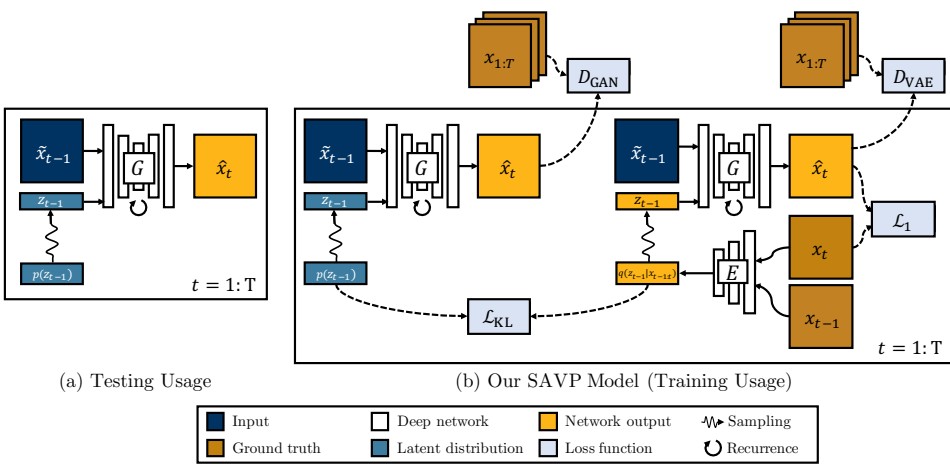

Figure 2: **Our proposed video prediction model.** (a) During testing, we synthesize new frames by sampling random latent codes $\mathbf{z}$ from a prior distribution $p(\mathbf{z})$ independently at each time step. The generator $G$ takes a previous frame and a latent code to synthesize a new frame. (b) During training, the generator is optimized to predict videos that match the distribution of real videos, using learned discriminators. The discriminators operate on entire sequences. We sample latent codes from two distributions: (1) the prior distribution, and (2) a posterior distribution approximated by a learned encoder $E$. For the latter, the regression $\mathcal{L}_1$ loss is used. Separate discriminators $D$ and $D^{\mathrm{VAE}}$ are used depending on the distribution used to sample the latent code.

During training, the latent code is sampled from $q(\mathbf{z}_t|\mathbf{x}_{t:t+1})$. The generation of each frame can be thought of as the reconstruction of frame $\hat{\mathbf{x}}_{t+1}$, where the ground truth frame $\mathbf{x}_{t+1}$ (along with $\mathbf{x}_t$) is encoded into a latent code $\mathbf{z}_t$, and then it (along with the last frame) is mapped back to $\hat{\mathbf{x}}_{t+1}$. Since the latent code has ground truth information about the frame being reconstructed, the model is encouraged to use it during training. This is a conditional version of VAEs, where the encoder and decoder are conditioned on the previous frame ($\mathbf{x}_t$ or $\hat{\mathbf{x}}_t$). To allow back-propagation through the encoder, the reconstruction term is rewritten using the re-parametrization trick,

$$\mathcal{L}_1(G, E) = \mathbb{E}_{\mathbf{x}_{0:T}, \mathbf{z}_t \sim E(\mathbf{x}_{t:t+1})|_{t=0}^{T-1}} \left[ \sum_{t=1}^{T} ||\mathbf{x}_t - G(\mathbf{x}_{t-1}, \mathbf{z}_{t-1})||_1 \right]. \tag{1}$$

To enable sampling from the prior at test time, a regularization term encourages the approximate posterior to be close to the prior distribution,

$$\mathcal{L}_{\mathrm{KL}}(E) = \mathbb{E}_{\mathbf{x}_{0:T}} \left[ \sum_{t=1}^{T} \mathcal{D}_{\mathrm{KL}}(E(\mathbf{x}_{t-1:t})||p(\mathbf{z}_{t-1})) \right]. \tag{2}$$

The VAE optimization involves minimizing the following objective, where the relative weighting of $\lambda_1$ and $\lambda_{\mathrm{KL}}$ is determined by the (fixed) variance of conditional likelihood $p(\mathbf{x}_t|\mathbf{x}_{t-1}, \mathbf{z}_{t-1})$,

$$G^*, E^* = \arg\min_{G, E} \lambda_1 \mathcal{L}_1(G, E) + \lambda_{\mathrm{KL}} \mathcal{L}_{\mathrm{KL}}(E). \tag{3}$$

## 3.2 GENERATIVE ADVERSARIAL NETWORKS

Without overcoming the problem of modeling pixel covariances, it is likely not possible to produce sharp and clean predictions. Indeed, as shown in our experiments, the pure VAE model tends to produce blurry futures. We can force the predictions to stay on the video manifold by matching the distributions of predicted and real videos. Given a classifier $D$ that is capable of distinguishing generated videos $\hat{\mathbf{x}}_{1:T}$ from real videos $\mathbf{x}_{1:T}$, the generator can be trained to match the statistics of the real data distribution using the binary cross-entropy loss,

$$\mathcal{L}_{\mathrm{GAN}}(G, D) = \mathbb{E}_{\mathbf{x}_{1:T}}[\log D(\mathbf{x}_{1:T})] + \mathbb{E}_{\mathbf{x}_{1:T}, \mathbf{z}_t \sim p(\mathbf{z}_t)|_{t=0}^{T-1}}[\log(1 - D(G(\mathbf{x}_0, \mathbf{z}_{0:T-1})))]. \tag{4}$$

The overloaded notation $G(\mathbf{x}_0, \mathbf{z}_{0:T-1})$ indicates the generated sequence $\hat{\mathbf{x}}_{1:T}$. The classifier, which is not known a priori and is problem-specific, can be realized as a deep discriminator network that can be adversarially learned,

$$G^* = \arg\min_{G} \max_{D} \mathcal{L}_{\mathrm{GAN}}(G, D). \tag{5}$$

This is the setting of GANs. In the conditional case, a per-pixel reconstruction term $\mathcal{L}_1^{\mathrm{GAN}}$ is added to the objective, which is analogous to $\mathcal{L}_1$, except that the latent codes are sampled from the prior.

### 3.3 Stochastic Adversarial Video Prediction

The VAE and GAN models provide complementary strengths. GANs use a learned loss function through the discriminator, which learns the statistics of natural videos. However, GANs can suffer from the problem of mode collapse, especially in the conditional setting (Pathak et al., 2016; Isola et al., 2017; Zhu et al., 2017). VAEs explicitly encourage the latent code to be more expressive and meaningful, since the learned encoder produces codes that are useful for making accurate predictions at training time. However, during training, VAEs only observe latent codes that are encodings of ground truth images, and never train on completely randomly drawn latent codes, leading to a potential train and test mismatch. GANs, however, are trained with randomly drawn codes.

Our stochastic adversarial video prediction (SAVP) model combines both approaches, shown in Fig. 2. Another term $\mathcal{L}_{\text{GAN}}^{\text{VAE}}$ is introduced, which is analogous to $\mathcal{L}_{\text{GAN}}$ except that it uses latent codes sampled from $q(\mathbf{z}_t|\mathbf{x}_{t:t+1})$ and a video discriminator $D^{\text{VAE}}$. The objective of our SAVP model is

$$G^*, E^* = \arg\min_{G,E} \max_{D,D^{\text{VAE}}} \lambda_1 \mathcal{L}_1(G,E) + \lambda_{\text{KL}} \mathcal{L}_{\text{KL}}(E) + \mathcal{L}_{\text{GAN}}(G,D) + \mathcal{L}_{\text{GAN}}^{\text{VAE}}(G,E,D^{\text{VAE}}). \quad (6)$$

### 3.4 Network Architectures

The generator is a convolutional LSTM (Xingjian et al., 2015) that predicts pixel-space transformations between the current and next frame, with additional skip connections with the first frame as done in SNA (Ebert et al., 2017). At every time step, the network is conditioned on the current frame and latent code. After the initial frames, the network is conditioned on its own predictions. The conditioning on the latent codes is realized by concatenating them along the channel dimension to the inputs of all the convolutional layers of the convolutional LSTM. We note that the warping component of this generator assumes that the frames in the videos can be described as transformations of pixels, which is the case for the datasets that we consider. And although the generator used in this work is based on SNA, any video generator (including the one from Denton & Fergus (2018)) could be used with our losses. The encoder is a feed-forward convolutional network that, at every time step, encodes a pair of images $\mathbf{x}_t$ and $\mathbf{x}_{t+1}$ into $\mu_{\mathbf{z}_t}$ and $\log \sigma_{\mathbf{z}_t}$.

The video discriminator is a feed-forward convolutional network with 3D filters, based on SNGAN (Miyato et al., 2018) but with the filters "inflated" from 2D to 3D. The network takes in a spatiotemporal cube of all the predicted pixels and outputs a single logit. The ground-truth context frames are not provided to the network. We found that spectral normalization in the discriminator and conditioning only on the predicted frames were important for a stable training. We also found that image discriminators that operate on single frames were not necessary. See Fig. 8 and Appendix A for additional details.

### 3.5 Discussion of Related VAE Models

Aside from the adversarial losses, the VAE component of our model is related to prior work on stochastic video prediction. Although the variational losses are the same, there are differences on encoding the posterior distribution and sampling the latent variables at training and test time.

The inference network of Babaeizadeh et al. (2018) estimates a single distribution $q(\mathbf{z}|\mathbf{x}_{1:T})$ by using a feed-forward network that encodes the entire video sequence at once. At test time, the latent variable is sampled from a unit Gaussian prior. They propose two variants for sampling. The latent is sampled once for the entire sequence in the time-invariant case or at every time step in the time-variant case.

On the other hand, the inference network of Denton & Fergus (2018) estimates a time-varying distribution $q(\mathbf{z}_t|\mathbf{x}_{1:t+1})$ by using a recurrent network that encodes all the frames up to the next frame. They propose two versions for the prior. The prior is either a fixed unit Gaussian distribution or a time-varying distribution $p(\mathbf{z}_t|\mathbf{x}_{1:t})$, which is learned and estimated by a recurrent network. In both cases, the latent variable is sampled at every time step.

In contrast, our inference network estimates a single distribution $q(\mathbf{z}_t|\mathbf{x}_{t:t+1})$ by using a feed-forward network that encodes the current and next frames. Unlike both prior works, the posterior distribution is temporally local and is conditioned on only two adjacent frames. The latent variables are sampled at every time step and, like the time-variant SV2P and fixed-prior SVG, the prior is a fixed unit Gaussian distribution for every time step.

## 4 EXPERIMENTS

Our experimental evaluation studies the realism, diversity, and accuracy of the videos generated by our approach and prior methods, and evaluates the importance of various design decisions, including the form of the reconstruction loss and the presence of the variational and adversarial objectives. Evaluating the performance of stochastic video prediction models is exceedingly challenging: not only should the samples from the model be physically realistic and visually plausible given the context frames, but the model should also be able to produce diverse samples that match the conditional distribution in the data. This is difficult to evaluate precisely: realism is not accurately reflected with simple metrics of reconstruction accuracy, and the true conditional distribution in the data is unknown, since real-world datasets only have a single future for each initial sequence. Below, we discuss the metrics that we use to evaluate realism, diversity, and accuracy. No single metric alone provides a clear answer as to which model is better, but considering multiple metrics can provide us with a more complete understanding of the performance and trade-offs of each approach.

### 4.1 EVALUATION METRICS

**Realism: comparisons to real videos using human judges.** The realism of the predicted videos is evaluated based on a *real vs. fake* two-alternative forced choice (2AFC) test. Human judges on Amazon Mechanical Turk (AMT) are presented with a pair of videos—one generated and one real—and asked to identify the generated, or "fake" video. We use the implementation from (Zhang et al., 2016), modified for videos. Each video is 10 frames long and shown over 2.5 seconds. For each method, we gather 1000 judgments from 25 human judges. Each human evaluator is provided with 10 training trials followed by 40 test trials. A method that produces perfectly realistic videos would achieve a fooling rate of 50%.

**Diversity: distance between samples.** Realism is not the only factor in determining the performance of a video prediction model: aside from generating predictions that look physically plausible and realistic, a successful model must also adequately cover the range of possible futures in an uncertain environment. We compute diversity as the average distance between randomly sampled video predictions, similar to Zhu et al. (2017). Distance is measured in the VGG feature space (pretrained on ImageNet classification), averaged across five layers, which has been shown to correlate well with human perception (Dosovitskiy & Brox, 2016; Johnson et al., 2016; Zhang et al., 2018).

**Accuracy: similarity of the best sample.** One weakness of the above metric is that the samples may be diverse but still not cover the feasible output space. Though we do not have the true output distribution, we can still leverage the single ground truth instance. This can be done by sampling the model a finite number of times, and evaluating the similarity between the best sample and the ground truth. This has been explored in prior work on stochastic video prediction (Babaeizadeh et al., 2018; Denton & Fergus, 2018), using PSNR or SSIM as the evaluation metric. In addition to these, we use cosine similarity in the pretrained VGG feature space.

### 4.2 DATASETS

We evaluate on two real-world datasets: the BAIR action-free robot pushing dataset (Ebert et al., 2017) and the KTH human actions dataset (Schuldt et al., 2004). See Appendix B.2 for additional results on the action-conditioned version of the robot pushing dataset.

**BAIR action-free.** This dataset consists of a randomly moving robotic arm that pushes objects on a table. This dataset is particularly challenging since (a) it contains large amounts of stochasticity due to random arm motion, and (b) it is a real-world application, with a diverse set of objects and large cluttered scene (rather than a single frame-centered object with a neutral background). The frame resolution is $64 \times 64$. We condition on 2 frames and train to predict the next 10 frames. We predict 10 future frames for the 2AFC experiments and 28 future frames for the other experiments.

**KTH.** This dataset consists of a human subject doing one of six activities: walking, jogging, running, boxing, hand waving, and hand clapping. For the first three activities, the human enters and leaves the frame multiple times, leaving the frame empty with a mostly static background for multiple frames at a time. The sequences are particularly stochastic when the initial frames are all empty since the human can enter the frame at any point in the future. As a preprocessing step, we center-crop each frame to a $120 \times 120$ square and then resize to a spatial resolution of $64 \times 64$. We condition on the first 10 frames and train to predict the next 10 frames. We predict 10 future

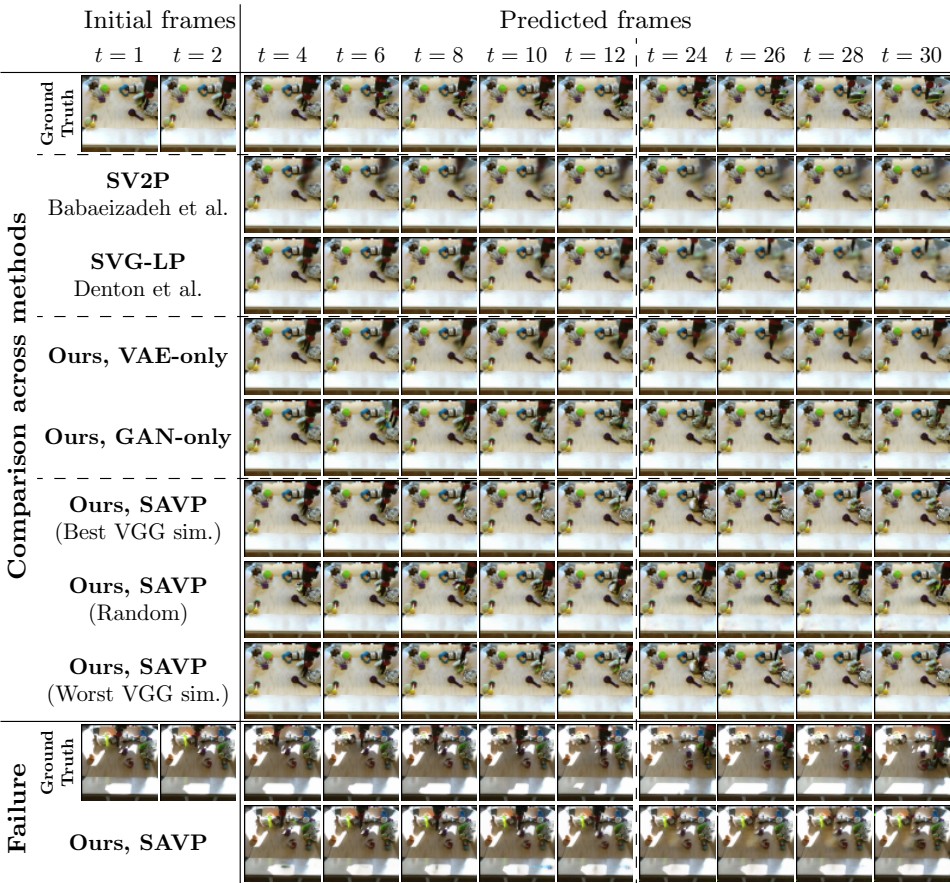

Figure 3: **Qualitative Results (BAIR action-free dataset).** Unless labeled otherwise, we show the closest generated sample to ground truth using VGG cosine similarity. SV2P immediately produces blurry results. Our GAN and VAE-based variants, as well as SVG-LP produce sharper results. However, SVG-LP still blurs out the jar on the right side of the image when it is touched by the robot, while our GAN-based models keep the jar sharp. We show three results for our SAVP model: using the closest, furthest, and random samples. There is large variation between the three samples in the arm motion, and even the furthest sample from the ground truth looks realistic. (bottom) We show a failure case where the arm disappears.

frames for the 2AFC experiments and 30 future frames for the other experiments. For each sequence, subclips of the desired length are randomly sampled at training and test time.

### 4.3 METHODS: ABLATIONS AND COMPARISONS

We compare the following variants of our method, in our effort to evaluate the effect of each loss term. Videos, code, and models are available at our website[1].

**Ours, SAVP.** Our stochastic adversarial video prediction model, with the VAE and GAN objectives.

**Ours, GAN-only.** An ablation of our model with only a conditional GAN, without the variational autoencoder. This model still takes a noise sample as input, but the noise is sampled from the prior during training. This model is broadly representative of prior stochastic GAN-based methods.

**Ours, VAE-only.** An ablation of our model with only a conditional VAE, with the reconstruction $\mathcal{L}_1$ loss but without the adversarial loss. This model is broadly representative of prior stochastic VAE-based methods.

**Ours, deterministic.** A deterministic ablation of our model with the reconstruction $\mathcal{L}_1$ loss but without the VAE nor the GAN objectives. The model uses the same generator architecture but without the latent variables.

---

[1]https://video-prediction.github.io/video_prediction

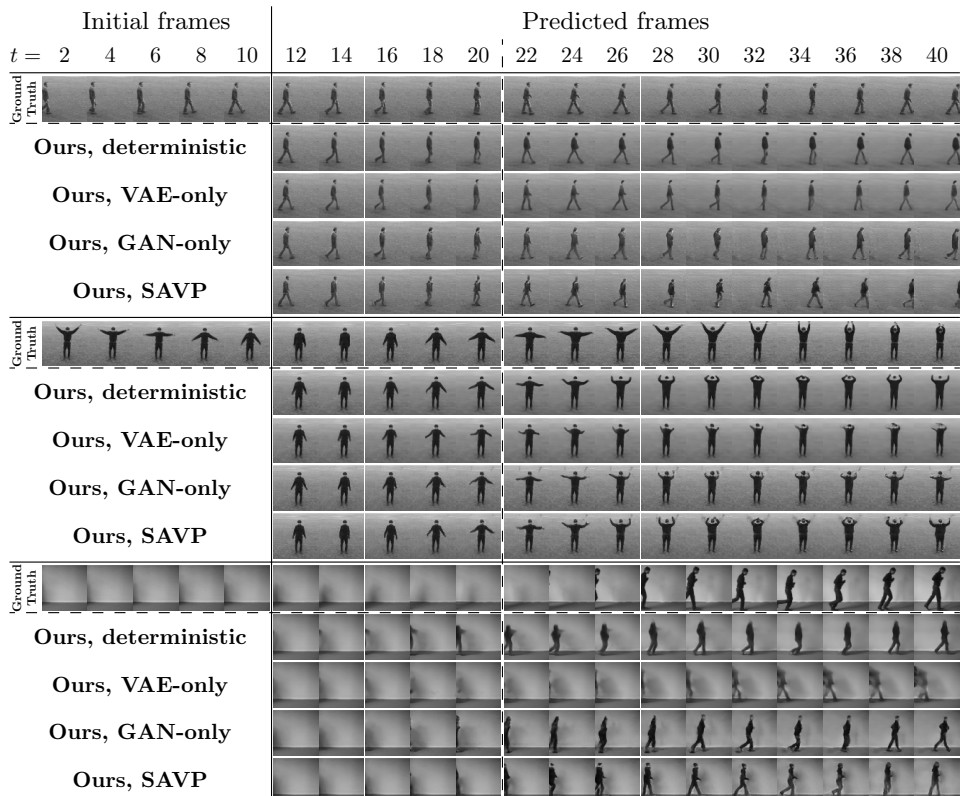

Figure 4: **Qualitative Results (KTH dataset).** We show qualitative comparisons to ablations of our model. Models are conditioned on 10 frames and trained to predict 10 future frames (vertical dashed line). For stochastic models, we show the closest generated sample to ground truth using VGG cosine similarity. We hypothesize that this dataset has much less stochasticity; even our deterministic model produces reasonable predictions. (top) Both the deterministic and VAE models generate images that are slightly blurry, but that do not degrade over time. The GAN-based methods produce sharper predictions. (middle) Our VAE model generates images where small limbs disappear further into the future, whereas our SAVP method preserves them. (bottom) All conditioning frames are empty except for a shadow on the left. All our variants are able to use this cue to predict that a person is coming from the left, although our SAVP model generates the most realistic sequence.

We also compare to prior stochastic VAE-based methods Stochastic Variational Video Prediction (SV2P) (Babaeizadeh et al., 2018) and Stochastic Video Generation (SVG) (Denton & Fergus, 2018), both of which use the reconstruction $\mathcal{L}_2$ loss and no adversarial loss.

## 4.4 EXPERIMENTAL RESULTS

We show qualitative results on the BAIR and KTH datasets in Fig. 3 and Fig. 4, respectively. For the quantitative results, we evaluate the realism, diversity, and accuracy of the predicted videos.

**Does our method produce *realistic* results?** In Fig. 5, variants of our method are compared to prior work. On the BAIR action-free dataset, our GAN variant achieves the highest fooling rate, whereas our proposed SAVP model, a VAE-GAN-based method, achieves a fooling rate that is roughly halfway between the GAN and VAE models alone. The SV2P method (Babaeizadeh et al., 2018) does not achieve realistic results. The VAE-based SVG-LP method (Denton & Fergus, 2018) achieves high realism, similar to our VAE variant, but substantially below our GAN-based variants. On the KTH dataset, our SAVP model achieves the highest realism score, substantially above our GAN variant. Among the VAE-based methods without adversarial losses, our VAE-only model outperforms SV2P and SVG-FP (Denton & Fergus, 2018) in terms of realism.

**Does our method generate *diverse* results?** We measure diversity by taking the distance between random samples. Diversity results are also shown in Fig. 5. For a qualitative visualization of diversity, see Appendix B.4. While the GAN-only approach achieves realistic results, it shows lower diversity than the VAE-based methods. This is an example of the commonly known phenomenon of

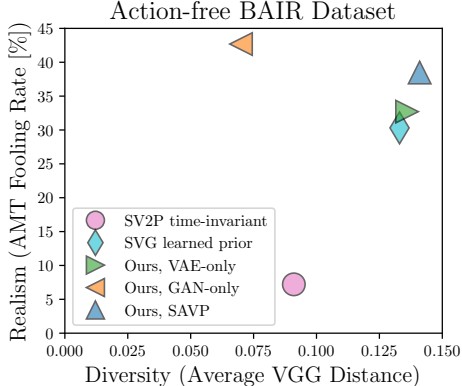 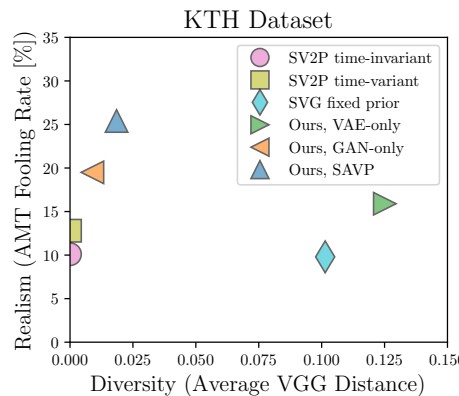

Figure 5: **Realism vs Diversity.** We measure realism using a real vs fake Amazon Mechanical Turk (AMT) test, and diversity using average VGG cosine distance. Higher is better on both metrics. **Our VAE** variant achieves higher realism and diversity than the **SV2P** (Babaeizadeh et al., 2018) and **SVG** (Denton & Fergus, 2018) methods based on VAEs. **Our GAN** variant achieves higher realism than the pure VAE methods, at the expense of significantly lower diversity. **Our SAVP** model, based on VAE-GANs, improves along the realism axis compared to a pure VAE method, and improves along the diversity axis compared to a pure GAN method. Although the SV2P methods mode-collapse on the KTH dataset, we note that they did not evaluate on this dataset, and their method could benefit from hyperparameters that are better suited for this dataset.

mode-collapse, where multiple latent codes produce the same or similar images on the output (Goodfellow, 2016). Intuitively, the VAE-based methods explicitly encourage the latent code to be more expressive by using an encoder from the output space into the latent space during training. This is verified in our experiments, as the VAE-based variants, including our SAVP model, achieve higher diversity than our GAN-only models on both datasets. On the KTH dataset, our VAE variant and VAE-based SVG-FP method (Denton & Fergus, 2018) both achieve significantly higher diversity than all the other methods. Although the VAE-based SV2P methods (Babaeizadeh et al., 2018) mode-collapse on the KTH dataset, we note that they did not evaluate on this dataset, and as such, their method could benefit from different hyperparameters that are better suited for this dataset.

**Does our method generate *accurate* results?** Following recent work on VAE-based video prediction (Babaeizadeh et al., 2018; Denton & Fergus, 2018), we evaluate on full-reference metrics by sampling multiple predictions from the model. We draw 100 samples for each video, find the "best" sample by computing similarity to the ground truth video, and show the average similarity across the test set as a function of time. The results on the BAIR and KTH datasets are shown in Fig. 14 and Fig. 15, respectively. We test generalization ability by running the model for more time steps than it was trained for. Even though the model is only trained to predict 10 future frames, we observe graceful degradation over time.

While PSNR and SSIM (Wang et al., 2004) are commonly used for video prediction, these metrics are not necessarily indicative of prediction quality. In video prediction, structural ambiguities and geometric deformations are a dominant factor, and SSIM is not an appropriate metric in such situations (Sampat et al., 2009; Zhang et al., 2018). This is particularly noticeable with the SV2P method, which achieves high PSNR and SSIM scores, but produces blurry and unrealistic images. Furthermore, we additionally trained our VAE and deterministic variants using the standard MSE loss $\mathcal{L}_2$ to understand the relationship between the form of the reconstruction loss and the metrics. The general trend is that models trained with $\mathcal{L}_2$, which favors blurry predictions, are better on PSNR and SSIM, but models trained with $\mathcal{L}_1$ are better on VGG cosine similarity. See Appendix B.1 for quantitative results comparing models trained with $\mathcal{L}_1$ and $\mathcal{L}_2$. In addition, we expect for our GAN-based variants to underperform on PSNR and SSIM since GANs prioritize matching joint distributions of pixels over per-pixel reconstruction accuracy.

To partially overcome the limitations of these metrics, we also evaluate using distances in a deep feature space (Dosovitskiy & Brox, 2016; Johnson et al., 2016), which have been shown to correspond better with human perceptual judgments (Zhang et al., 2018). We use cosine similarity between VGG features averaged across five layers. Otherwise, a model trained for it would unfairly and artificially achieve better similarities by exploiting potential flaws on that metric. Our VAE variant, along with SVG (Denton & Fergus, 2018), performs best on this metric. Although our SAVP model

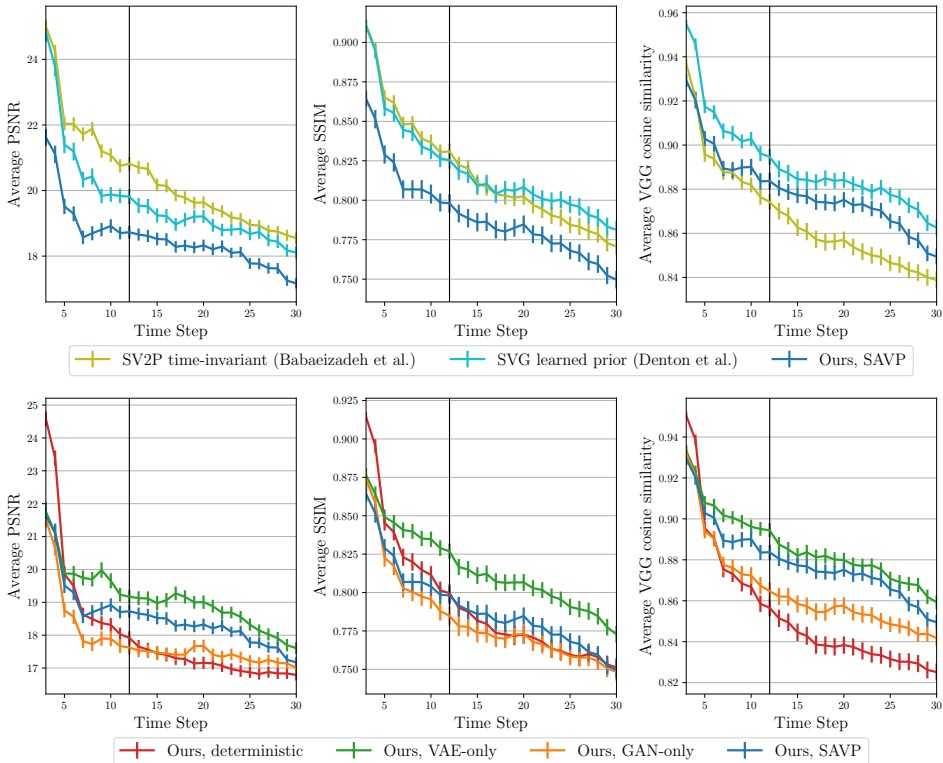

Figure 6: **Similarity of the best sample (BAIR action-free dataset).** We show the similarity (higher is better) between the best sample (of 100) as a function of prediction time step across different methods and evaluation metrics. (top) Although SV2P produces blurry and unrealistic images, it achieves the highest PSNR. Both SAVP and SVG-LP outperform SV2P on VGG similarity. We expect our GAN-based variants to underperform on PSNR and SSIM since GANs prioritize matching joint distributions of pixels over per-pixel reconstruction accuracy. (bottom) We compare to ablated versions of our model. Our VAE variant achieves higher scores than our SAVP model, which in turn achieves significantly higher VGG similarities compared to our GAN-only model. Note that the models were only trained to predict 10 future frames (indicated by the vertical line), but is being tested on generalization to longer sequences.

improves on diversity and realism, it also performs worse in accuracy compared to pure VAE models (both our own ablation and SVG). This is to be expected, since accuracy and realism are at odds with each other. This tradeoff has recently been proved and it holds even for similarity distances based on VGG features (Blau & Michaeli; Blau et al., 2018). Among the VAE-based methods, SV2P (Babaeizadeh et al., 2018) achieves the lowest VGG similarity.

On stochastic environments, such as in the BAIR action-free dataset, there is correlation between diversity and accuracy of the best sample: a model with diverse predictions is more likely to sample a video that is close to the ground truth. This relation can be seen in Fig. 5 and Fig. 14 for the robot dataset, e.g. our SAVP model is both more diverse and achieves higher similarity than our GAN-only variant. This is not true on less stochastic environments. We hypothesize that the KTH dataset is not as stochastic when conditioning on 10 frames, as evidenced the similarities between the predictions from the deterministic and stochastic models. This would explain why our GAN variant and SV2P achieve modest similarities despite achieving low diversity on the KTH dataset.

**Does combining the VAE and GAN produce better predictions?** The GAN alone achieves high realism but low diversity. The VAE alone achieves lower realism but increased diversity. Adding the GAN to the VAE model increases the realism without sacrificing diversity, at only a small or no cost in realism on stochastic datasets. This is consistent with Zhu et al. (2017), which showed that combining GAN and VAE-based models provides benefits in the case of image generation. To our knowledge, our method is the first to extend this class of models to the video prediction setting, and the first to illustrate that this leads to improved realism with a degree of diversity comparable to the best VAE models in stochastic environments. The results show that this combination of losses is the best choice for realistic coverage of diverse stochastic futures.

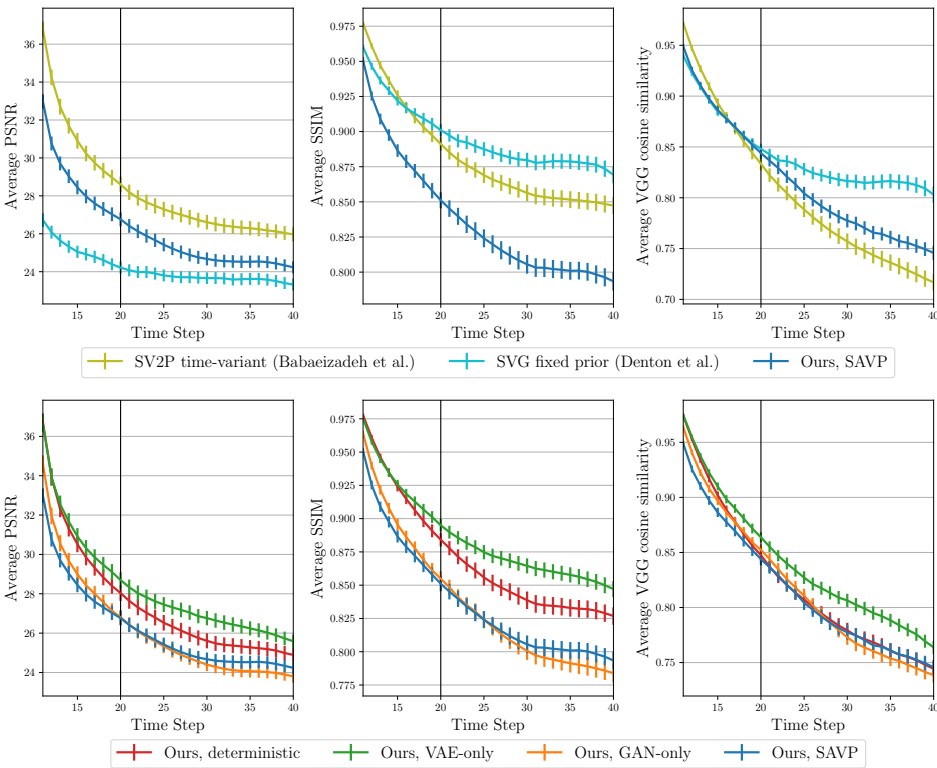

Figure 7: **Similarity of the best sample (KTH dataset).** We evaluate the similarity between the best predicted sample (out of a 100 samples) and the ground truth video. (top) As in the case of the robot dataset, SV2P achieves high PSNR values, even though it produces blurry and unrealistic images. Although all three methods achieve comparable VGG similarities for the first 10 future frames (which is what the models were trained for, and indicated by the vertical line), our SAVP model predicts videos that are substantially more realistic, as shown in our subjective human evaluation, thus achieving a desirable balance between realism and accuracy. (bottom) We compare to ablated versions of our model. Our VAE-only method outperforms all our other variants on the three metrics. In addition, our deterministic model is not that far behind in terms of similarity, leading us to believe that the KTH dataset is not as stochastic when conditioning on the past 10 frames.

## 5 CONCLUSION

We develop a video prediction model that combines latent variables trained via a variational lower bound with an adversarial loss to produce a high degree of visual and physical realism. VAE-style training enables our method to make diverse stochastic predictions, and our experiments show that the adversarial loss is effective at producing predictions that are more visually realistic according to human raters. Evaluation of video prediction models is a major challenge, and we evaluate our method, as well as ablated variants that consist of only the VAE or only the GAN loss, in terms of a variety of quantitative and qualitative measures, including human ratings, diversity, and accuracy of the predicted samples. Our results demonstrate that our approach produces more realistic predictions than prior methods, while preserving the sample diversity of VAE-based methods.

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

# A NETWORKS AND TRAINING DETAILS

## A.1 NETWORK DETAILS

### A.1.1 GENERATOR.

Our generator network, shown in Fig. 8, is inspired by the convolutional dynamic neural advection (CDNA) model proposed by Finn et al. (2016). The video prediction setting is a sequential prediction problem, so we use a convolutional LSTM (Hochreiter & Schmidhuber, 1997; Xingjian et al., 2015) to predict future frames. We initialize the prediction on the initial sequence of ground truth frames (2 or 10 frames for the BAIR and KTH datasets, respectively), and predict 10 future frames. The model predicts a sequence of future frames by repeatedly making next-frame predictions and feeding those predictions back to itself. For each one-step prediction, the predicted frame is given by a compositing layer, which composes intermediate frames with predicted compositing masks. The intermediate frames include the previous frame, transformed versions of the previous frame, and a frame with pixels directly synthesized by the network. The transformed versions of the frame are produced by convolving in the input image with predicted convolutional kernels, allowing for different shifted versions of the input. In more recent work, the first frame of the sequence is also given as one of the intermediate frames (Ebert et al., 2017).

To enable stochastic sampling, the generator is also conditioned on time-varying latent codes, which are sampled at training and test time. Each latent code $\mathbf{z}_t$ is an 8-dimensional vector. At each prediction step, the latent code is passed through a fully-connected LSTM to facilitate correlations in time of the latent variables. The encoded latent code is then passed to all the convolutional layers of the main network, by concatenating it along the channel dimension to the inputs of these layers. Since they are vectors with no spatial dimensions, they are replicated spatially to match the spatial dimensions of the inputs.

We made a variety of architectural improvements to the original CDNA (Finn et al., 2016) and SNA (Ebert et al., 2017) models, which overall produced better results on the per-pixel loss and similarity metrics. See Fig. 11 for a quantitative comparison of our deterministic variant (without

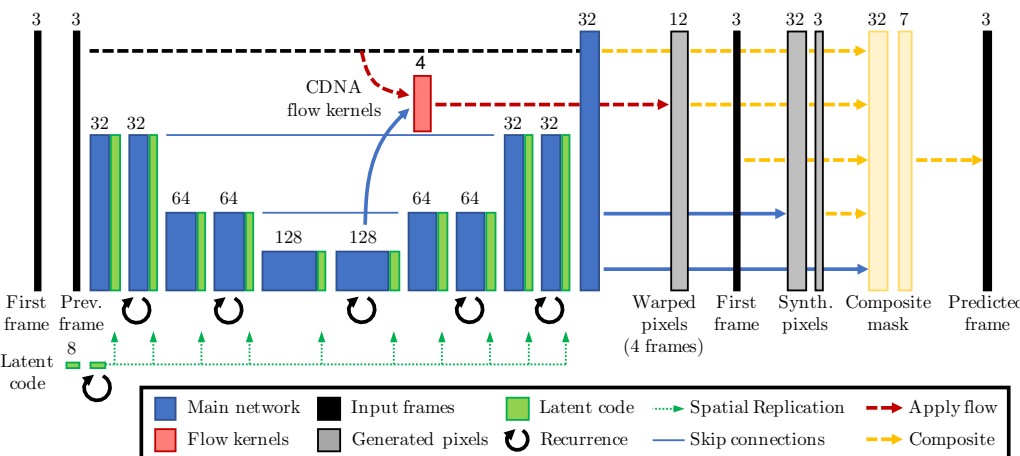

Figure 8: **Architecture of our generator network.** Our network uses a convolutional LSTM (Hochreiter & Schmidhuber, 1997; Xingjian et al., 2015) with skip-connection between internal layers. As proposed by Finn et al. (2016), the network predicts (1) a set of convolution kernels to produce a set of transformed input images (2) synthesized pixels at the input resolution and (3) a compositing mask. Using the mask, the network can choose how to composite together the set of warped pixels, the first frame, previous frame, and synthesized pixels. One of the internal feature maps is given to a fully-connected layer to compute the kernels that specify pixel flow. The output of the main network is passed to two separate heads, each with two convolutional layers, to predict the synthesized frame and the composite mask. These two outputs use sigmoid and softmax non-linearities, respectively, to ensure proper normalization. We enable stochastic sampling of the model by conditioning the generator network on latent codes. These are first passed through a fully-connected LSTM, and then given to all the convolutional layers of the the convolutional LSTM.

VAE or GAN losses) to the SNA model on the BAIR action-conditioned dataset. Each convolutional layer is followed by instance normalization (Ulyanov et al., 2016) and ReLU activations. We also use instance normalization on the LSTM pre-activations (i.e., the input, forget, and output gates, as well as the transformed and next cell of the LSTM). In addition, we modify the spatial downsampling and upsampling mechanisms. Standard subsampling and upsampling between convolutions is known to produce artifacts for dense image generation tasks (Odena et al., 2016; Zhao et al., 2017; Niklaus et al., 2017). In the encoding layers, we reduce the spatial resolution of the feature maps by average pooling, and in the decoding layers, we increase the resolution by using bilinear interpolation. All convolutions in the generator use a stride of 1. In the case of the action-conditioned dataset, actions are concatenated to the inputs of all the convolutional layers of the main network, as opposed to only the bottleneck.

### A.1.2 Encoder.

The encoder is a standard convolutional network that, at every time step, encodes a pair of images $\mathbf{x}_t$ and $\mathbf{x}_{t+1}$ into $\mu_{\mathbf{z}_t}$ and $\log \sigma_{\mathbf{z}_t}$. The latent variable $\mathbf{z}_t$ is sampled at every time step and the same encoder network with shared weights is used at every step. The encoder architecture consists of three convolutional layers, followed by average pooling of all the spatial dimensions. Two separate fully-connected layers are then used to estimate $\mu_{\mathbf{z}_t}$ and $\log \sigma_{\mathbf{z}_t}$, respectively. The convolutional layers use instance normalization, leaky ReLU non-linearities, and stride 2. This encoder architecture is the same one used in BicyleGAN (Zhu et al., 2017) except that the inputs are pair of images, concatenated along the channel dimension.

### A.1.3 Discriminator.

The discriminator is a 3D convolutional neural network that takes in all the images of the video at once. We use spectral normalization and the SNGAN discriminator architecture (Miyato et al., 2018), except that we "inflate" the convolution filters from 2D to 3D. The two video discriminators, $D$ and $D_{\mathrm{VAE}}$, share the same architecture, but not the weights, as done in BicycleGAN (Zhu et al., 2017).

## A.2 Training Details

Our generator network uses scheduled sampling during training as in Finn et al. (2016), such that at the beginning the model is trained for one-step predictions, while by the end of training the model is fully autoregressive. We trained all models with Adam (Kingma & Ba, 2015) for 300000 iterations, linearly decaying the learning rate to 0 for the last 100000 iterations. The same training schedule was used for all the models, except for SVG, which was trained by its author. Our GAN-based variants used an optimizer with $\beta_1 = 0.5$, $\beta_2 = 0.999$, learning rate of 0.0002, and a batch size of 16. Our deterministic and VAE models (including SNA and SV2P from prior work) used an optimizer with $\beta_1 = 0.9$, $\beta_2 = 0.999$, learning rate of 0.001, and a batch size of 32.

We used $\lambda_1 = 100$ for our GAN-based variants, and $\lambda_1 = 1$ for all the other models. For our VAE-based variants, we linearly anneal the weight on the KL divergence term from 0 to the final value $\lambda_{\mathrm{KL}}$ during training, as proposed by Bowman et al. (2016), from iterations 50000 to 100000. We used a relative weighting of $\lambda_{\mathrm{KL}}/\lambda_1 = 0.001$ for the BAIR robot pushing datasets, and $\lambda_{\mathrm{KL}}/\lambda_1 = 0.00001$ for the KTH dataset. This hyperparameter was empirically chosen by computing similarity metrics on the validation set.

## B Additional Experiments

### B.1 Comparison of Pixel-Wise $\mathcal{L}_1$ and $\mathcal{L}_2$ Losses

We train our deterministic and VAE variants with the $\mathcal{L}_1$ and $\mathcal{L}_2$ losses to compare the effects of these reconstruction losses on the full-reference metrics used in this work. The pixel-wise $\mathcal{L}_1$ loss assumes that pixels are generated according to a fully factorized Laplacian distribution, whereas the $\mathcal{L}_2$ loss corresponds to a fully factorized Gaussian distribution. See Fig. 9 and Fig. 11 for quantitative results on the action-free and action-conditioned BAIR datasets, respectively.

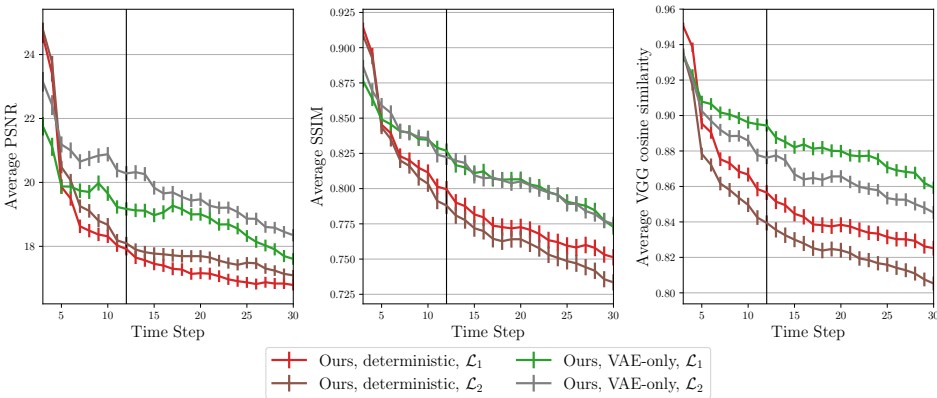

Figure 9: **Similarity of the best sample (BAIR action-conditioned dataset).** We show the similarity between the predicted video and the ground truth, using the same evaluation as in Fig. 14. We compare our deterministic and VAE variants when trained with $\mathcal{L}_1$ and $\mathcal{L}_2$ losses, and observe that they have a significant impact on the quality of our predictions. The models trained with $\mathcal{L}_1$ produce videos that are qualitatively better and achieve higher VGG similarity than the equivalent models trained with $\mathcal{L}_2$.

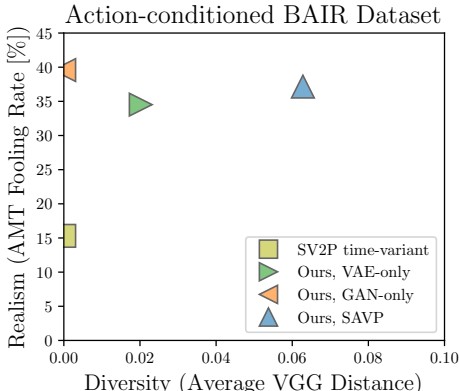

Figure 10: **Realism vs Diversity (BAIR action-conditioned dataset).** The SV2P method (Babaeizadeh et al., 2018) from prior work produces images with low realism, whereas our GAN, VAE, and SAVP models fool the human judges at a rate of around 35-40%. Our VAE-based models also produce videos with higher diversity, though lower diversity than other datasets, as this task involves much less stochasticity. The trend is the same as in the other datasets. Our SAVP model improves the realism of the predictions compared to our VAE-only model, and improves the diversity compared to our GAN-only model.

The general trend is that models trained with the $\mathcal{L}_2$ loss tend to generate blurry predictions but achieve higher PSNR scores than equivalent models trained with the $\mathcal{L}_1$ loss. This is because PSNR and $\mathcal{L}_2$ are closely related, the former being a logarithmic function of the latter. The opposite is true for the VGG cosine similarity metric, which has been shown to correspond better with human perceptual judgments (Zhang et al., 2018). Models trained with $\mathcal{L}_1$ significantly outperforms equivalent models trained with $\mathcal{L}_2$. On the SSIM metric, models trained with $\mathcal{L}_1$ achieve roughly the same or better similarities than models trained with $\mathcal{L}_2$. Although both losses are pixel-wise losses, the choice between $\mathcal{L}_1$ and $\mathcal{L}_2$ have a significant impact on the quality of our predicted videos.

## B.2 RESULTS ON ACTION-CONDITIONED BAIR ROBOT PUSHING DATASET

We use the same dataset as the one in the main paper, except that we use the robot actions. Each action is a 4-dimensional vector corresponding to Cartesian translations and a value indicating if the gripper has been closed or opened. As in the action-free dataset, we condition on the first 2 frames of the sequence and train to predict the next 10 frames. In this dataset, the video prediction model is now also conditioned on a sequence of actions $\mathbf{a}_{0:T-1}$, in addition to the initial frames. The

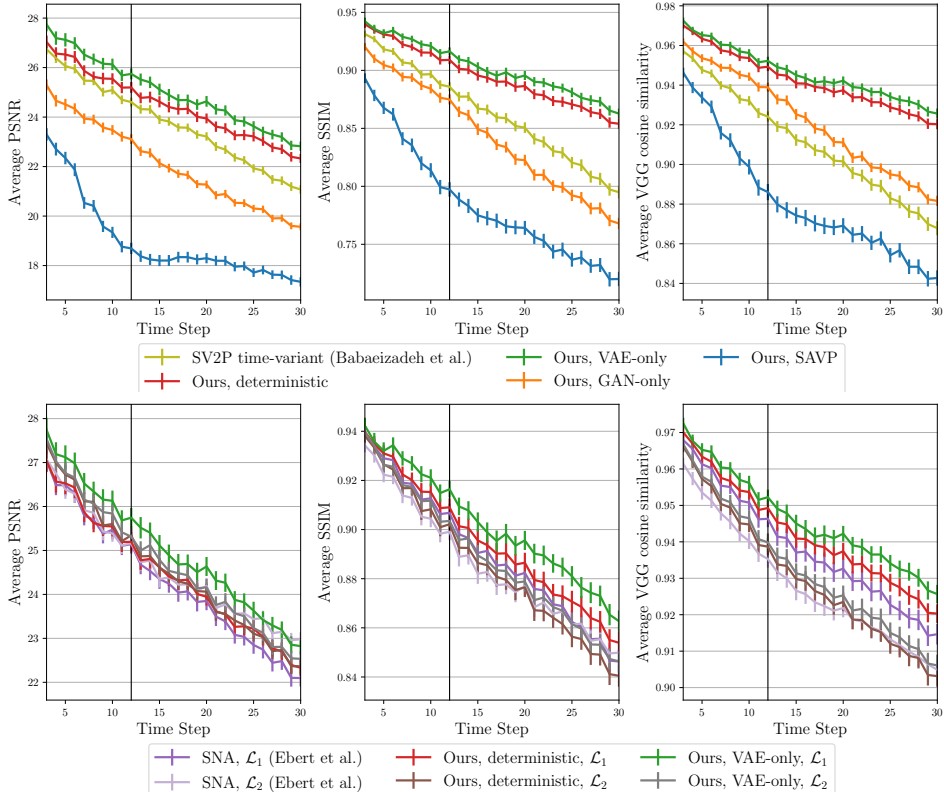

Figure 11: **Similarity of the best sample (BAIR action-conditioned dataset).** We show the similarity between the predicted video and the ground truth, using the same evaluation as in Fig. 14, except that we condition on robot actions. (top) We compare to prior SV2P (Babaeizadeh et al., 2018) and ours ablations. Our VAE and deterministic models both outperform SV2P, even though it is VAE-based. However, notice that the gap in performance between our VAE and deterministic models is small, as the dataset is less stochastic when conditioning on actions. Our SAVP model achieves much lower scores on all three metrics. We hypothesize that our SAVP model, as well as SV2P, is underutilizing the provided actions and thus achieving more stochasticity at the expense of accuracy. (bottom) We compare deterministic models—SNA (Ebert et al., 2017) and ours— and our VAE model when trained with $\mathcal{L}_1$ and $\mathcal{L}_2$ losses. As in the action-free case, we observe that the choice of the pixel-wise reconstruction loss significantly affects prediction accuracy. Models trained with $\mathcal{L}_1$ are substantially better in SSIM and VGG cosine similarity compared to equivalent models trained with $\mathcal{L}_2$. Surprisingly, the VAE model trained with $\mathcal{L}_1$ outperforms the other models even on the PSNR metric. We hypothesize that VAE models trained with $\mathcal{L}_1$ are better equipped to separate multiple modes of futures, whereas the ones trained with $\mathcal{L}_2$ might still average some of the modes. In fact, we evidenced this in preliminary experiments on the toy shapes dataset used by Babaeizadeh et al. (2018). Among the deterministic models, ours improves upon SNA (Ebert et al., 2017), which is currently the best deterministic action-conditioned model on this dataset.

generator network is modified to take an action $\mathbf{a}_t$ at each time step, by concatenating the action to the inputs of all the convolutional layers of the main network, similar to how the latent code $\mathbf{z}_t$ is passed in (but without the additional fully-connected LSTM).

We show the realism and diversity results in Fig. 10 and the accuracy results in Fig. 11. In addition to the methods compared in the action-free dataset, we also compare to SNA (Ebert et al., 2017), an action-conditioned deterministic video prediction model. The results indicate that our VAE model significantly outperforms prior methods on the full-reference metrics, and that our models significantly outperforms the model by Babaeizadeh et al. (2018) both in terms of diversity of predictions and realism.

|  | Generated frames | | | | | | | | |
|---|---|---|---|---|---|---|---|---|---|
|  | $t=1$ | $t=3$ | $t=5$ | $t=7$ | $t=9$ | $t=11$ | $t=13$ | $t=15$ | $t=17$ | $t=19$ |
| **MoCoGAN** **(Patch)** Tulyakov et al. | 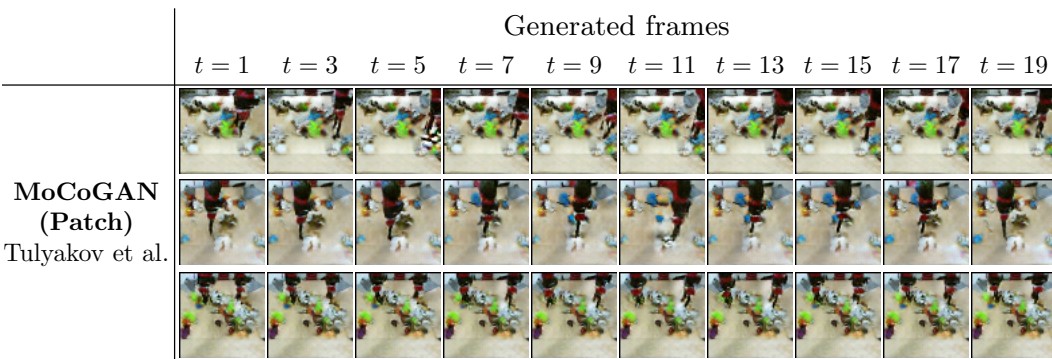 | | | | | | | | | |

Figure 12: **Example generations of MoCoGAN.** We use the unconditional version of MoCoGAN (Tulyakov et al., 2018) to generate videos from the BAIR robot pushing dataset. We chose this model as a representative recent example of purely GAN-based unconditioned video generation. MoCoGAN produces impressive results on various applications related to human action, which are focused on a actor in the middle of the frame. However, this model struggles in the robot dataset where multiple entities are moving at a time. Note that since the patch-based discriminator has a limited receptive field of the image, the model can produce videos with two robot arms (last row) even though this is not in the dataset. We did not observe this behavior with their image-based discriminator.

## B.3 Example Generations of an Unconditional GAN

We consider the motion and content decomposed GAN (MoCoGAN) model (Tulyakov et al., 2018) as a representative unconditional GAN method from prior work, and use it to generate videos from the BAIR robot pushing dataset. We use their publicly available code and show qualitative results in Fig. 12. The results show the variant that uses patch-based discriminators, since this one achieved higher realism than the variant that uses image-based discriminators. Since this prior work demonstrates competitive results in comparison to other prior unconditional GAN methods, we chose it as the most representative recent example of purely GAN-based video generation for this comparison. MoCoGAN produces impressive results on various applications related to human action, which are focused on a single actor in the middle of the frame. However, it struggles on videos in the robot pushing domain where multiple entities are moving at a time, i.e. the robot arm and the objects it interacts with.

## B.4 Qualitative visualization of diversity

We show a qualitative visualization of diversity in Fig. 13 by averaging multiple samples.

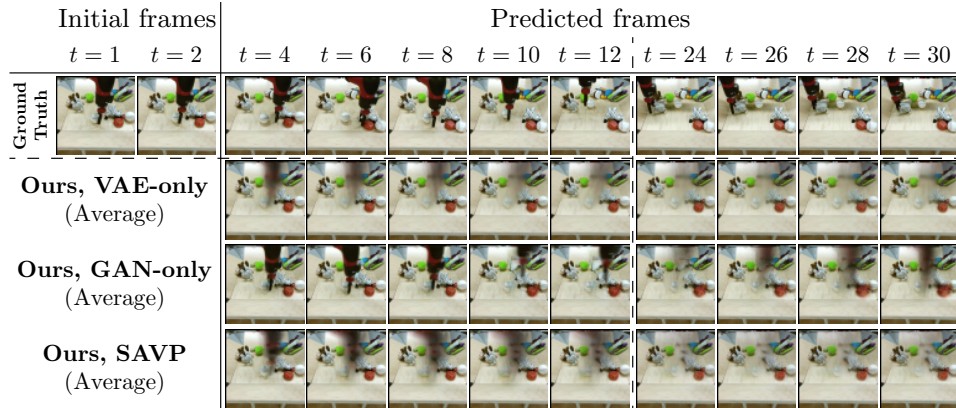

Figure 13: **Qualitative visualization of diversity.** We show predictions of our models, averaged over 100 samples. A model that produces diverse outputs should predict that the robot arm moves in random directions at each time step, and thus the arm should "disappear" over time in these averaged predictions. Consistent with our quantitative evaluation of diversity, we see that both our SAVP model and our VAE variant produces diverse samples, whereas the GAN-only method is prone to mode-collapse.

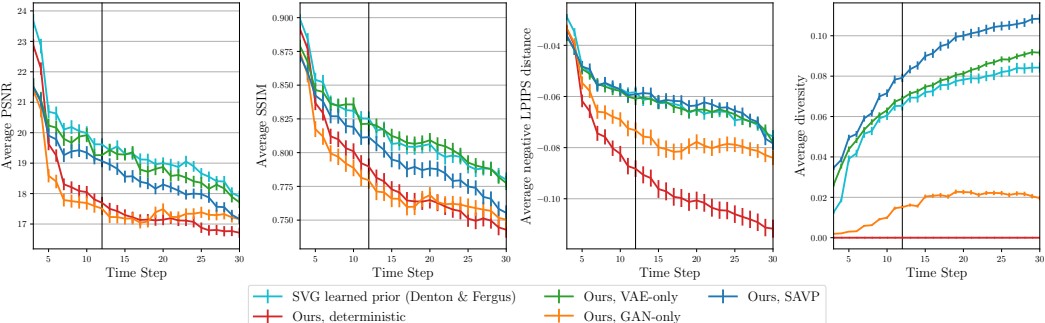

Figure 14: **Similarity of the best sample (BAIR action-free dataset), updated plot.** We show the similarity (higher is better) between the best sample (of 100) as a function of prediction time step across different methods and evaluation metrics. The three leftmost plots show the similarity (higher is better) between the best sample (of 100) as a function of prediction time step across different methods and evaluation metrics. Besides the standard metrics, we also use the Learned Perceptual Image Patch Similarity (LPIPS) metric (Zhang et al., 2018), which has been shown to correlate well with human perception. This distance is measured in the AlexNet feature space (pretrained on ImageNet classification) with linear weights calibrated to match human judgements. The plot on the right shows the diversity (higher is better) as a function of prediction time step, computed as the LPIPS distance between pairs of samples. Aside for the first two predicted frames, our SAVP model achieves similar LPIPS distances as the VAE models, both our VAE ablation and the SVG model from Denton & Fergus (2018). In addition, not only our SAVP method substantially improve sample diversity compared to the GAN-only model, but it also produces more diverse samples than both of the VAE models.

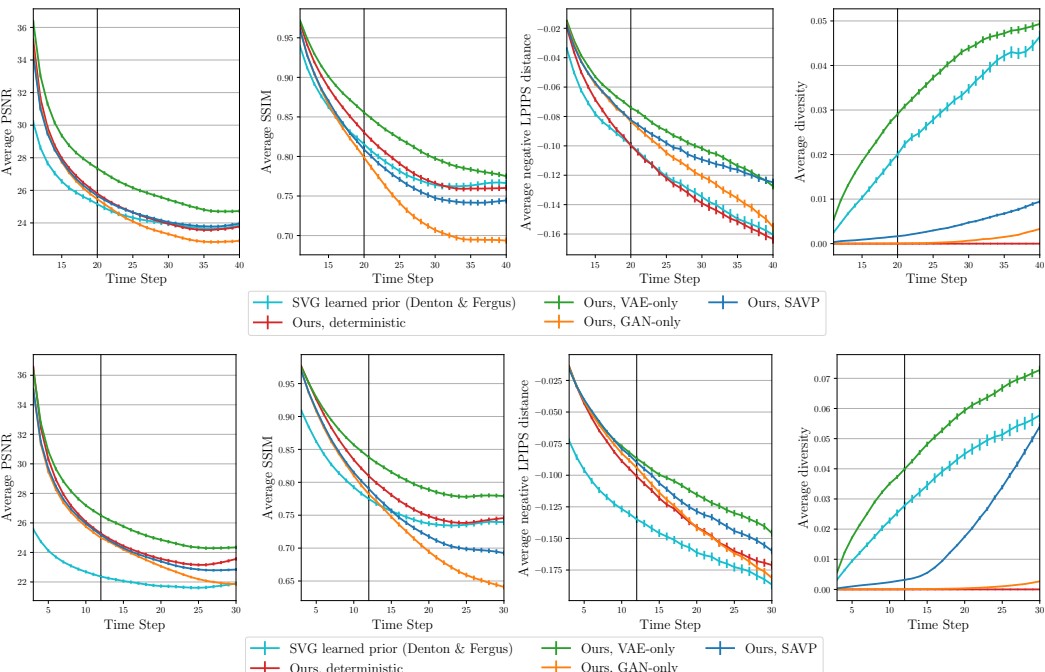

Figure 15: **Similarity of the best sample and diversity (KTH dataset), updated plot.** The three leftmost plots show the similarity (higher is better) between the best sample (of 100) as a function of prediction time step. The plot on the right shows the diversity (higher is better) as a function of prediction time step, computed as the LPIPS distance between pairs of samples. The top and bottom plots show results when conditioning on 10 and 2 frames, respectively. Among the VAE methods, our VAE-only model achieves substantially higher similarities and diversities than the SVG model from prior work (Denton & Fergus, 2018). The GAN-only model mode-collapses and generates samples that lack diversity. Our SAVP method, which incorporates the variational loss, improves both sample diversity and similarities, compared to the GAN-only model. Our SAVP model also achieves higher accuracy than SVG.

