# OpenReview forum: "Stochastic Adversarial Video Prediction"
_ICLR.cc/2019/Conference_

### Official Review · AnonReviewer1 · 2018-10-31
**VAE-GAN model for video prediction**

**Rating:** 5
**Confidence:** 5

**Review:**


Summary:
The authors present a video prediction model called SAVP that combines a Variational Auto-Encoder (VAE) model with a Generative Adversarial Network (GAN) to produce more realistic and diverse future samples.

Deterministic models and certain loss functions such as Mean Squared Error (MSE) will produce
blurry results when making uncertain predictions. GAN predictions on the other hand usually are more visually appealing but often lack diversity, producing just a few modes. The authors propose to combine a VAE model with a GAN objective to combine their strengths: good quality samples (GAN) that cover multiple possible futures (VAE).

Strengths:
[+] GANs are notoriously unstable to train, especially for video. The authors formulate a VAE-GAN model and successfully implement it.

Weaknesses:
[-] The combination of VAEs and GANs, while new for videos, had already been proposed for image generation as indicated in the Related Work section and its formulation for video prediction is relatively straightforward given existing VAE (Denton & Fergus 2018) and GAN models (Tulyakov et al. 2018).

[-] The results indicate that SAVP offers a trade-off between the properties of GANs and VAEs, but does not go beyond its individual parts. For example, the experiment of Figure 5 does not show SAVP being significantly more diverse than GANs for KTH (as compared to VAEs). Furthermore, Figure 6 and Figure 7 in general show SAVP performing worse than SVG (Denton & Fergus 2018), a VAE model with a significantly less complex generator, including for the metric (VGG cosine similarity) that the authors introduce arguing that PSNR and SSIM do not necessarily indicate prediction quality.

While the use of a GAN in general will make the results less blurry and visually appealing, it does not necessarily mean that the samples it generates are going to be plausible or better. Since a direct application of video prediction is model-based planning, it seems that plausibility might be as important as sample quality. This work proposes to combine VAEs and GANs in a single model to get the benefits of both models. However, the experiments conducted generally show that SAVP offers only a trade-off between the visual quality of GANs and the coverage of VAEs, and does not show a clear advantage over current VAE models (Denton & Fergus, 2018) that with simpler architectures obtain similar results. While the presentation is clear and the evaluation of the model is thorough, I am unsure of the significance of the proposed method.

In order to better assess this model and compare it to its individual parts and other VAE models, could the authors:

1) Compare SAVP to the SVG-LP/FP model on a controlled synthetic dataset such as Stochastic Moving MNIST (Denton & Fergus, 2018)?
2) Comment on the plausibility of the samples generated by SAVP? Do some samples show imagined objects – implausible interactions for the robotic arm dataset? If so, what would be the advantage over blurry but plausible generations of a VAE?

---

> ### Author Response · Authors · 2018-11-22
> **We address the questions and clarify a few details, which are also reflected in the updated draft**
>
> We thank reviewer 1 for the detailed feedback. In this response, we clarify the accuracy-realism trade-off, revise the accuracy metrics, indicate reruns and new experiments, and address the individual questions.
>
> We updated Section 4.4 to indicate that it is to be expected that, although our SAVP model improves on diversity and realism, it also performs worse in accuracy compared to pure VAE models (both our own ablation and SVG). A recent result [1] proves that there is a fundamental tradeoff between accuracy and realism, for all problems with inherent ambiguity. In fact, a recent challenge held at ECCV 2018 in such a problem [2] evaluates all algorithms on both of these axes, as neither adequately captures performance.
>
> Although the SVG generator is simpler than ours, ours is just a simple variation from Ebert et al. (2017). Since proposing a strong generator architecture is not the goal of this paper,
> any video generator (including the one from Denton & Fergus (2018)) could be used with our losses. We added this clarification to Section 3.4. Instead, we provide a systematic analysis of the effect of the loss function on this task (which could be applied to any generator). It's also worth noting that with a simpler feed-forward posterior and a unit Gaussian prior, our VAE ablation and SVG achieve similar performance on various metrics. We added Section 3.5 to point out the differences between the VAE component of our model and prior work.
>
> We have included a revised plot in Figure 14 (note that this temporary plot will be incorporated into Figure 6), where we use the official implementation of SSIM and replace the VGG metric with the LPIPS metric (Zhang et al., 2018). LPIPS linearly calibrates AlexNet feature space to better match human perceptual similarity judgements. Aside for the first two predicted frames, our VAE ablation and the SVG model both achieve similar SSIM and LPIPS.
>
> After examining the KTH results further, we realized that our results are likely weaker than they should have been, because we did not use the same preprocessing as prior work. The experiments from our original submission cropped the videos into a square before resizing, and thus discarded information from the sides of the video. We are currently rerunning the KTH experiments and we plan to update the results in the paper. We also didn't choose particular hyperparameters to ensure diversity for our models, and we expect some improvement in diversity in the new sets of experiments.
>
> Although the combination of VAEs and GANs have been explored recently for conditional image generation (Zhang et al. 2018), the video prediction task is substantially different, with unique challenges, due to spatiotemporal relationships and inherent compounding uncertainty of the future.
>
> Furthermore, while the individual components have indeed been known for video prediction, their combination is novel and not present in prior work, and we demonstrate that this produces state-of-the-art results in terms of diversity and realism. In addition, this work provides a detailed comparison of the effect of the losses on the various metrics. Furthermore, we are currently running experiments for various weightings of the KL loss and the adversarial loss, and we plan to include additional results that illustrate the trade-offs based on these hyperparameters.
>
> Although MoCoGAN performs well for videos with a single frame-centered actor, it struggles with multiple simultaneously moving entities. The authors of MoCoGAN also mentioned in personal correspondence that the conditional version (i.e. video prediction) was significantly harder to train. We noticed the same in earlier iterations of our model. In our case, we found that the model would degenerate to static videos or videos with a cyclic flickering artifact, which are issues that aren't a problem in conditional image generation. We added details to Section 3.4 describing the importance of a few components, such as spectral normalization and not conditioning the discriminator in the ground-truth context frames.
>
> The purpose of adding adversarial losses to a pure VAE is to improve on blurry predictions where the latent variables alone cannot capture the uncertainty of the data. However, that is typically not the case of synthetic datasets. In early experiments, we trained our pure VAE model on the stochastic shape movement dataset from Babaeizadeh et al. (2018), and our pure VAE was able to model the dataset without any blur and with perfect separation of the possible futures.
>
> We agree that plausibility is indeed important, and that's what our human subject studies try to capture. Since we provide predictions of the whole sequence to the human evaluator, we are not only evaluating for image realism but also for plausibility of the dynamics. Unlike the VAE models that implausibly erase the small objects that are being pushed in the BAIR dataset, our SAVP model moves those objects in a more plausible way.

---

> > ### Author Response · Authors · 2018-11-22
> > **References for the previous post**
> >
> > [1] Yochai Blau and Tomer Michaeli. The perception-distortion tradeoff. In Conference on Vision and Pattern Recognition (CVPR), 2018. https://arxiv.org/abs/1711.06077
> >
> > [2] Yochai Blau, Roey Mechrez, Radu Timofte, Tomer Michaeli, and Lihi Zelnik-Manor. 2018 PIRM Challenge on Perceptual Image Super-resolution. In Perceptual Image Restoration and Manipulation (PIRM) workshop at ECCV 2018. https://arxiv.org/abs/1809.07517

---

> > ### Comment · AnonReviewer1 · 2018-11-25
> > **Requests not fully addressed**
> >
> > ------------------------------------------------
> > * The purpose of adding adversarial losses to a pure VAE is to improve on blurry predictions where the latent variables alone cannot capture the uncertainty of the data. However, that is typically not the case of synthetic datasets. In early experiments, we trained our pure VAE model on the stochastic shape movement dataset from Babaeizadeh et al. (2018), and our pure VAE was able to model the dataset without any blur and with perfect separation of the possible futures.
> >
> > While it's true that both are synthetic datasets, there are two significant differences between Stochastic Moving MNIST and the stochastic shape movement dataset from Babaeizadeh et al. (2018) - in Moving MNIST the objects have a greater variety of shapes as they are digits and not simple polygons, and in Moving MNIST the crossings between the digits are significantly hard to model because of the uncertainty in the resulting shape. Most VAE models including SVG struggle to not produce blurry frames when digits cross. Given this, I still believe that, if indeed the proposed SAVP model removes the blurriness of VAE predictions while still producing plausible interactions, showing the performance of SAVP on Stochastic Moving MNIST would be a useful experiment that would make it easier to evaluate the model - a middle ground between too simple toy tasks and more complex datasets.
> >
> > ------------------------------------------------
> > * We agree that plausibility is indeed important, and that's what our human subject studies try to capture. Since we provide predictions of the whole sequence to the human evaluator, we are not only evaluating for image realism but also for plausibility of the dynamics. Unlike the VAE models that implausibly erase the small objects that are being pushed in the BAIR dataset, our SAVP model moves those objects in a more plausible way
> >
> > My experience is that video GAN models for the BAIR Pushing Dataset don't produce blurry results but instead they imagine new objects/shapes when there are object interactions. Usually these predictions look 'more natural', specially for low resolution predictions (64x64 pixels), but upon closer inspection it is easy to spot new objects, implausible motions, etc. My original comment was meant to ask whether this still happens with SAVP or the combination of VAE + GAN solves this issue. In my opinion this is an important question - imagine we want to use a video prediction model for planning. Then, it is unclear to me whether it would be better to use blurry but physically sound predictions (VAE) compared to sharper predictions with imagined objects/interactions (GAN). While I understand that the user study was meant to answer this question, I believe regular users would prefer the sharper predictions even when some of the interactions are not physically possible. My original comment was meant to ask the authors whether SAVP doesn't suffer from these imagined objects and interactions.

---

> > > ### Author Response · Authors · 2018-11-27
> > > **Revised plot for the KTH dataset shows that our method substantially outperform prior SVG method in terms of accuracy, and responses to other comments.**
> > >
> > > We have included a revised plot in Figure 15 at the end of the Appendix (which will be incorporated to Figure 7) that fixes the KTH dataset preprocessing. Our VAE-only model now achieves substantially higher accuracy and diversity than SVG (Denton & Fergus, 2018). As before, the GAN-only model mode-collapses and generates samples that lack diversity. Our SAVP method, which incorporates the variational loss, improves both sample diversity and similarities, compared to the GAN-only model. Our SAVP model also achieves higher accuracy than SVG. The experiments from our original submission (1) cropped the videos into a square before resizing, and thus discarded information from the sides of the video, and (2) did not filter out the empty frames, and thus our models were trained on uninformative frames. We fixed those issues to match the preprocessing used by Denton & Fergus (2018). In addition, we have also included experiments where we condition on only 2 frames instead of 10 frames, in order to test on a setting with more stochasticity.
> > >
> > > We thank the reviewer for clarifying the challenges of the Moving MNIST dataset. We are currently running experiments on that dataset, but unfortunately they won't be done by the rebuttal deadline. We will update the website once we have those results ready.
> > >
> > > Regarding the second comment, we expect the GAN and the SVAP models to produce qualitatively similar images. We uploaded the predictions of the test set to the supplementary anonymous website [3]. In our predictions, we have noticed implausible object motions in the form of object deformations (e.g. samples 4, 6) or sometimes transformation into new objects (e.g. samples 16, 35). A pure GAN and our SAVP method produce roughly equally plausible images, as evidenced by the realism results in Figure 5. The main gain of the variational loss with respect to a pure GAN is in the diversity of the samples.
> > >
> > > When these models are used in downstream tasks, it depends on the exact nature of the downstream task whether blurry but physically sound predictions are preferred over sharper predictions that occasionally produces imagined interactions. For example, a visual servoing controller might benefit from smoother and blurry predictions, whereas a controller that uses a goal-image classifier might benefit from predictions that are in the image manifold. Our paper (a) contributes to the Pareto frontier in this space and (b) characterizes this tradeoff for future reference.
> > >
> > > [3] https://video-prediction.github.io/video_prediction/index_files/tables/bair_action_free_all/index.html

---

### Official Review · AnonReviewer2 · 2018-11-02
**Straightforward paper and simple extension of VAE-GANs**

**Rating:** 6
**Confidence:** 4

**Review:**

This paper proposes to extend VAE-GAN from the static image generation setting to the video generation setting. It’s a well-written, simple paper that capitalizes on the trade-off between model realism and diversity, and the fact that VAEs and GANs (at least empirically) tend to lie on different sides of this spectrum.

The idea to extend the use of VAE-GANs to the video prediction setting is a pretty natural one and not especially novel. However, the effort to implement it successfully is commendable and will, I think, serve as a good reference for future work on video prediction.

There are also several interesting design choices that I think are worth of further exposition. Why, for example, did the authors only perform variational inference with the current and previous frames? Did conditioning on additional frames offer limited further improvement? Can the blurriness instead be attributable to the weak inference model? Please provide a response to these questions. If the authors have any ablation studies to back up their design choices, that would also be much appreciated, and will make this a more valuable paper for readers.

I think Figure 5 is the most interesting figure in the paper. I would imagine that playing with the hyperparameters would allow one to traverse the trade-off between realism and diversity. I think having such a curve will help sell the paper as giving the practitioner the freedom to select their own preferred trade-off.

I don’t understand the claim that “GANs prioritize matching joint distributions of pixels over per-pixel reconstruction” and its implication that VAEs do not prioritize joint distribution matching. VAEs prioritize matching joint distributions of pixels and latent space: min KL(q(z, x) || p(z, x)) and is a variational approximation of the problem min KL(q(x) || p(x)), where q(x) is the data distribution. The explanation provided by the authors is thus not sufficiently precise and I recommend the retraction of this claim.

Pros:
+ Well-written
+ Natural extension of VAE-GANs to video prediction setting
+ Establishes a good baseline for future video prediction work
Cons:
- Limited novelty
- Limited analysis of model/architecture design choices

---

> ### Author Response · Authors · 2018-11-21
> **We address the questions and add clarifications, which are also reflected in the updated draft**
>
> We thank reviewer 2 for the detailed feedback. We are glad that the reviewer found the VAE-GAN model to be a natural extension for the problem and that our work provides a good baseline for future work. We address the individual questions below.
>
> We changed Section 3.1 to explain that the posterior dependence on pairs of adjacent frames is to have temporally local latent variables that capture the ambiguity for only that transition, a sensible choice when using i.i.d. Gaussian priors. Another choice is to use temporally correlated latent variables, which would require a stronger prior (e.g. as in Denton & Fergus (2018)). For simplicity, we opted for the former.
>
> The blurriness in a VAE can indeed be attributable to a weak inference model. Note that our VAE variant and both SVG variants are able to predict sharp robot arms in the BAIR dataset, but often blur out the small objects being pushed. We tried recurrent posteriors and learned priors with our models, and the results were similar. We are now running additional experiments with a deeper encoder and with more filters. Although in principle a strong inference model could produce sharper images, an alternative approach is to use better losses, which is the approach we chose in this work.
>
> It is an interesting suggestion to experiment with the effect of the hyperparameters on the trade-off between realism and diversity. We are currently running experiments for various weightings of the KL loss and the adversarial loss, and we plan to include results that illustrate the trade-offs based on these hyperparameters. We also plan to include results on the trade-offs between accuracy and realism. In fact, a recent result [1] proves that this is a fundamental trade-off for all problems with inherent ambiguity.
>
> The statement that “GANs prioritize matching joint distributions of pixels over per-pixel reconstruction" is a criticism of per-pixel losses, and not of VAEs in general. We clarified in the introduction that VAEs can indeed model joint distributions of pixels.
>
> [1] Yochai Blau and Tomer Michaeli. The perception-distortion tradeoff. In Conference on Vision and Pattern Recognition (CVPR), 2018. https://arxiv.org/abs/1711.06077

---

### Official Review · AnonReviewer3 · 2018-11-02
**the paper proposes an extension of VAE based video prediction models and produces an extensive evaluation. The model seems to perform well, the originality and the improvement w.r.t. baselines are somewhat limited.**

**Rating:** 6
**Confidence:** 3

**Review:**

The paper introduces a generative model for video prediction. The originality stems from a new training criterion which combines a VAE and a GAN criteria. At training time, the GAN and the VAE are trained simultaneously with a shared generator; at test time, prediction conditioned on initial frames is performed by sampling from a latent distribution and generating the next frames via an enhanced conv LST . Evaluations are performed on two movement video datasets classically used for benchmarking  this task - several quantitative evaluation criteria are considered.

The paper clearly states the objective and provides a nice general description of the method.  The proposed model extends previous work by adding an adversarial loss to a VAE video prediction model.  The evaluation compares different variants of this model to two recent VAE baselines. A special emphasis is put on the quantitative evaluation: several criteria are introduced for characterizing different properties of the models with a focus on diversity. w.r.t. the baselines, the model behaves well for the “realistic” and “diversity” measures. The results are more mitigated for measures of accuracy. As for the qualitative evaluation, the model corrects the blurring effect of the reference SV2P baseline, and produces quite realistic predictions on these datasets. The difference with the other reference model (SVG) is less clear.

While the general description of the model is clear, details are lacking. It would probably help to position the VAE component more precisely w.r.t. one of the two baselines, by indicating the differences. This would also help to explain the difference of performance/ behavior  w.r.t. these models (Fig. 5).

It seems that the discriminator takes a whole sequence as input, but some precision on how this done could be added.  Similarly, you did not indicate what the deterministic version of your model is.
The generator model with its warping component makes a strong hypothesis on the nature of the videos: it seems especially well suited for translations or for other simple geometric transformations characteristics of the benchmarking videos .  Could you comment on the importance of this component? Did you test the model on other types of videos where this hypothesis is less relevant? It seems that the baseline SVG makes use of simpler ConLSTM for example.

The description of the generator in the appendix is difficult to follow. I missed the point in the following sentence: “For each one-step prediction, the network has the freedom to choose to copy pixels from the previous frame, used transformed versions of the previous frame, or to synthesize pixels from scratch” .
Also, it is not clear from the discussion on z, whether sampling is performed once for each video of for each frame.

Overall, the paper proposes an extension of VAE based video prediction models and produces an extensive evaluation. While the model seems to perform well, the originality and the improvement w.r.t. baselines are somewhat limited.

---

> ### Author Response · Authors · 2018-11-21
> **We address the questions and add clarifications and details to the paper**
>
> We thank reviewer 3 for the detailed feedback. We are glad that the reviewer found the extensive evaluation appropriate, and that our model behaves well for the realistic and diversity measures. We now address all the individual questions.
>
> We added Section 3.5 to point out the differences between the VAE component of our model and the SV2P and SVG models from prior work. In Section 3.4, we clarified what frames the discriminator takes, and in Section 4.3 we added a description of the deterministic version of our model. In Section A.1.1, we provided a better description of how frames are predicted at each time step. In Section 3.5 and A.1.2, we clarified that the latent variables are sampled at every time step.
>
> We updated Section 4.4 to indicate that it is to be expected that although our SAVP model improves on diversity and realism, it also performs worse in accuracy compared to pure VAE models (both our own ablation and SVG from Denton & Fergus (2018)). A recent result [1] proves that there is a fundamental tradeoff between accuracy and realism, for all problems with inherent ambiguity. In fact, a recent challenge held at ECCV 2018 in such a problem [2] evaluates all algorithms on both of these axes, as neither adequately captures performance.
>
> Note that proposing a generator architecture is not the goal of this paper. Instead, we provide a systematic analysis of the effect of the loss function on this task (which could be applied to any generator). We use a warping-based generator, from prior work (Ebert et al. 2017), and include a comparison to SVG for completeness. In the updated draft, we clarify in Section 3.4 that the warping component assumes that videos can be described as transformation of pixels, but that any generator (including the one from Denton & Fergus (2018)) could be used with our losses. Since evaluating generator architectures is not the emphasis of this paper, we did not test the importance of the warping component nor test on videos where this hypothesis is less suitable.
>
> We have included a revised plot in Figure 14 at the end of the Appendix (note that this temporary plot will be incorporated to Figure 6), where we use the official implementation of SSIM and replace the VGG metric with the Learned Perceptual Image Patch Similarity (LPIPS) metric (Zhang et al., 2018).  LPIPS linearly calibrates  AlexNet feature space to better match human perceptual similarity judgements. Aside for the first two predicted frames, our VAE ablation and the SVG model both achieve similar SSIM and LPIPS performance.
>
> [1] Yochai Blau and Tomer Michaeli. The perception-distortion tradeoff. In Conference on Vision and Pattern Recognition (CVPR), 2018. https://arxiv.org/abs/1711.06077
>
> [2] Yochai Blau, Roey Mechrez, Radu Timofte, Tomer Michaeli, and Lihi Zelnik-Manor. 2018 PIRM Challenge on Perceptual Image Super-resolution. In Perceptual Image Restoration and Manipulation (PIRM) workshop at ECCV 2018. https://arxiv.org/abs/1809.07517

---

> > ### Author Response · Authors · 2018-11-27
> > **Revised plot for the KTH dataset shows that our method substantially outperform prior SVG method in terms of accuracy.**
> >
> > We have included a revised plot in Figure 15 at the end of the Appendix (which will be incorporated to Figure 7) that fixes the KTH dataset preprocessing. Our VAE-only model now achieves substantially higher accuracy and diversity than SVG (Denton & Fergus, 2018). As before, the GAN-only model mode-collapses and generates samples that lack diversity. Our SAVP method, which incorporates the variational loss, improves both sample diversity and similarities, compared to the GAN-only model. Our SAVP model also achieves higher accuracy than SVG. The experiments from our original submission (1) cropped the videos into a square before resizing, and thus discarded information from the sides of the video, and (2) did not filter out the empty frames, and thus our models were trained on uninformative frames. We fixed those issues to match the preprocessing used by Denton & Fergus (2018). In addition, we have also included experiments where we condition on only 2 frames instead of 10 frames, in order to test on a setting with more stochasticity.

---

### Author Response · Authors · 2018-11-27
**Revised plot (Fig. 15) for the KTH dataset shows that our method substantially outperform prior SVG method in terms of accuracy.**

We have included a revised plot in Figure 15 at the end of the Appendix (which will be incorporated to Figure 7) that fixes the KTH dataset preprocessing. Our VAE-only model now achieves substantially higher accuracy and diversity than SVG (Denton & Fergus, 2018). As before, the GAN-only model mode-collapses and generates samples that lack diversity. Our SAVP method, which incorporates the variational loss, improves both sample diversity and similarities, compared to the GAN-only model. Our SAVP model also achieves higher accuracy than SVG. The experiments from our original submission (1) cropped the videos into a square before resizing, and thus discarded information from the sides of the video, and (2) did not filter out the empty frames, and thus our models were trained on uninformative frames. We fixed those issues to match the preprocessing used by Denton & Fergus (2018). In addition, we have also included experiments where we condition on only 2 frames instead of 10 frames, in order to test on a setting with more stochasticity.

---

### Author Response · Authors · 2018-11-27
**Revised plot (Fig. 14) for the BAIR action-free dataset shows that our method achieves similar LPIPS accuracy and better diversity than the VAE models, including prior SVG method.**

We have updated the revised plot in Figure 14 to include updated metrics from all of our methods evaluated on the BAIR action-free dataset.  Aside for the first two predicted frames, our SAVP model achieves similar LPIPS distances as the VAE models, both our VAE ablation and the SVG model from Denton & Fergus (2018). In addition, not only our SAVP method substantially improve sample diversity compared to the GAN-only model, but it also produces more diverse samples than both of the VAE models. We conducted a coarse search over hyperparameters on the validation set, this time varying the weightings of the KL divergence and the GAN loss with values 1, 0.1, and 0.01. We found that a lower weighting of 0.1 (instead of 1) for the GAN loss led to better accuracy. Unfortunately, we did not have time to rerun the user study to evaluate on realism, but we plan to do it in a future revision. Qualitatively, the predictions look at least as realistic as before.

---

### Meta-Review · Area_Chair1 · 2018-12-16
**Significance of applying a GAN - VAE combination to video is too limited**

**Confidence:** 4
**Recommendation:** Reject

**Metareview:**

This paper shows that combining GAN and VAE for video prediction allows to trade off diversity and realism. The paper is well-written and the experimentation is careful, as noted by reviewers. However, reviewers agree that this combination is of limited novelty (having been used for images before). Reviewers also note that the empirical performance is not very much stronger than baselines. Overall, the novelty is too slight and the empirical results are not strong enough compared to baselines to justify acceptance based solely on empirical results.